# The Identification and Gene Mapping of Spotted Leaf Mutant *spl43* in Rice

**DOI:** 10.3390/ijms25126637

**Published:** 2024-06-17

**Authors:** Chen Wang, Wen-Jun Liu, Xin-Wei Liao, Xia Xu, Shihua Yang, Xiao-Bo Zhang, Hai Zhou, Chuxiong Zhuang, Junyi Gong, Jian-Li Wu

**Affiliations:** 1State Key Laboratory of Rice Biology, China National Rice Research Institute, Hangzhou 310006, China; 984810137wc@gmail.com (C.W.); wenjunl413@163.com (W.-J.L.); myapple_pig@sina.com (X.-W.L.); mailxuxia@163.com (X.X.); yangshihua6308@163.com (S.Y.); zhangxiaobo@caas.cn (X.-B.Z.); 2State Key Laboratory for Conservation and Utilization of Subtropical Agro-Bioresources, South China Agricultural University, Guangzhou 510642, China; haizhou@scau.edu.cn (H.Z.); zhuangcx@scau.edu.cn (C.Z.)

**Keywords:** spotted leaf mutant, *OsRPT5A*, map-based cloning, cell death, defense response, reactive oxygen species

## Abstract

Our study investigates the genetic mechanisms underlying the spotted leaf phenotype in rice, focusing on the *spl43* mutant. This mutant is characterized by persistent reddish-brown leaf spots from the seedling stage to maturity, leading to extensive leaf necrosis. Using map-based cloning, we localized the responsible locus to a 330 Kb region on chromosome 2. We identified *LOC_Os02g56000*, named *OsRPT5A*, as the causative gene. A point mutation in *OsRPT5A*, substituting valine for glutamic acid, was identified as the critical factor for the phenotype. Functional complementation and the generation of CRISPR/Cas9-mediated knockout lines in the IR64 background confirmed the central role of OsRPT5A in controlling this trait. The qPCR results from different parts of the rice plant revealed that *OsRPT5A* is constitutively expressed across various tissues, with its subcellular localization unaffected by the mutation. Notably, we observed an abnormal accumulation of reactive oxygen species (ROS) in *spl43* mutants by examining the physiological indexes of leaves, suggesting a disruption in the ROS system. Complementation studies indicated OsRPT5A’s involvement in ROS homeostasis and catalase activity regulation. Moreover, the *spl43* mutant exhibited enhanced resistance to *Xanthomonas oryzae* pv. *oryzae* (*Xoo*), highlighting OsRPT5A’s role in rice pathogen resistance mechanisms. Overall, our results suggest that OsRPT5A plays a critical role in regulating ROS homeostasis and enhancing pathogen resistance in rice.

## 1. Introduction

Rice (*Oryza sativa* L.) is a vital staple food crop globally, yet its productivity is frequently compromised by various biotic stresses, including attacks by pathogens. Understanding the genetic mechanisms behind these stresses is crucial for developing disease-resistant rice varieties [1].

The spotted leaf mutant is a suitable platform for understanding the resistance of rice to pests and diseases. Spotted leaf mutants (spl-mutants) in rice are characterized by spontaneous lesion formation on leaves, mirroring disease-like symptoms in the absence of pathogens. These mutants serve as a valuable tool for unraveling the molecular mechanisms underlying plant immunity and programmed cell death (PCD) [2], and they hold promise for breeding disease-resistant rice varieties.

As a tool, spl-mutants display a range of phenotypic traits, including the spontaneous emergence of necrotic lesions on leaves, which vary in size, shape, and distribution depending on the specific mutant. For instance, the *blm* mutant features large, spreading lesions and is linked to enhanced resistance to blast pathogens, suggesting a potential role in defense [3]. Temperature and light conditions can also influence lesion formation in spl-mutants. The *lrd* mutant, for instance, shows increased lesion formation under high light intensity, highlighting the interaction between environmental factors and genetic predisposition in lesion development [4].

Recent advancements in genetic mapping and molecular cloning have unveiled several genes responsible for spotted leaf phenotypes in rice. These genes play pivotal roles in a variety of biological processes, including PCD, defense signaling, and stress responses. The *SPL11* gene, which encodes a U-box/armadillo repeat protein with E3 ubiquitin ligase activity, negatively regulates cell death and defense mechanisms. This suggests its involvement in the ubiquitination-mediated degradation of proteins associated with defense [5]. The *OsSPL1* gene, encoding a sphingosine-1-phosphate lyase, is a key player in sphingolipid metabolism and is associated with disease resistance responses [6]. Furthermore, some spl-mutant genes are crucial for the regulation of reactive oxygen species (ROS) production and signaling. For instance, the *OsLMS* gene, which encodes a protein with double-stranded RNA-binding motifs, is implicated in RNA metabolism and ROS accumulation. This leads to lesion formation and early senescence [7]. The interaction between various hormonal pathways, such as salicylic acid (SA), jasmonic acid (JA), and ethylene (ET), is also vital in modulating the spotted leaf phenotype. The *OsEDR1* gene, for example, negatively regulates bacterial resistance by activating ethylene biosynthesis, demonstrating the intricate crosstalk among hormone signaling pathways in plant immunity [8].

Investigations into rice spotted leaf mutants have shed light on the intricate molecular mechanisms governing plant immunity and PCD. Unraveling the genetic basis of spl-mutants paves the way for crafting rice varieties with enhanced disease resistance, thereby bolstering sustainable agriculture and ensuring food security. Here, we selected and characterized a spotted leaf mutant from the IR64 mutant library. The mutant *spl43* exhibited excessive accumulation of H_2_O_2_ on leaves, which might result in the occurrence of the spotted leaf phenotype. The phenotype was controlled by the single recessive nuclear gene *LOC_Os02g56000*. The mutation of *LOC_Os02g56000* enhanced the resistance to *Xoo* in rice.

## 2. Results

### 2.1. The Mutation of OsRPT5A Leads to the Spotted Leaf Phenotype

In our rice spotted leaf mutant library, we focused particularly on a mutant named *spl43*, which exhibited reddish-brown spots from the seedling stage, persisting throughout the plant’s life cycle, and ultimately leading to extensive leaf necrosis (Figure 1A,B and Appendix A). To elucidate the genetic mechanism underlying this phenotype, we employed a map-based cloning approach and analyzed the progeny of a cross between *spl43* and the japonica rice variety 80A90YR72. We selected individual plants exhibiting the mutant phenotype from the F_2_ population for mapping. Specifically, 111 plants displayed the mutant phenotype, while 489 plants exhibited the wild-type phenotype. Initially, genomic DNA from eight individual mutant plants was randomly selected to create a pooled sample. Primers distributed across the 12 rice chromosomes were screened, showing polymorphisms between IR64 and the japonica rice variety 80A90YR72. Subsequently, 30 individual plants with the mutant phenotype were selected, and approximately 200 SSR markers evenly distributed across the 12 rice chromosomes were used to locate the gene controlling the spotted leaf trait (Appendix A). Preliminary mapping positioned this gene between markers RM250 and D240B on chromosome 2. To further refine the mapping, additional markers such as C154-2, C154-11, and C154-27 were designed. Utilizing 80 individual plants with the mutant phenotype, we narrowed the interval to a 330 Kb region between markers C154-11 and C154-27.

Through a comprehensive analysis of related literature and targeted sequencing of candidate genes within the narrowed locus, we identified the target gene *LOC_Os02g56000*. Given its high similarity to the Arabidopsis gene *RPT5A*, we named it *OsRPT5A*. This discovery was significant as a point mutation from thymine (T) to adenine (A) occurred in the eighth exon of *OsRPT5A*, leading to the substitution of valine (Val) for glutamic acid (Glu) in the protein, which is likely a key factor causing the spotted leaf phenotype (Figure 1C).

To further validate this hypothesis, we conducted functional complementation experiments in the *spl43* mutant background and successfully obtained transgenic lines with restored phenotypes, strongly confirming the crucial role of *OsRPT5A* in controlling the spotted leaf trait. Moreover, we generated *OsRPT5A* knockout lines in the IR64 background, resulting in positive lines Cr-1 and Cr-2 that exhibited the spotted leaf phenotype, indicating that the loss of *OsRPT5A* can lead to the spotted leaf phenotype (Figure 1D,E).

The leaf color of the *spl*-mutant has changed locally. To verify whether the mutation affects the normal growth and development of the plant, we measured the chlorophyll content of the transgenic plants. We found that the chlorophyll content of the complementation lines was close to that of the wild type, while the knockout lines were similar to the mutant (Figure 1F). The agronomic traits of the knockout lines were close to *spl43*, and some agronomic traits of the complementation lines were restored to the wild-type level (Appendix A). This implies that the presence of low levels of various photosynthetic pigments in the mutant leads to defects in the natural processes of light absorption, energy transfer, and the capture of additional light energy in the leaves. Consequently, this results in insufficient accumulation of organic matter, explaining why the agronomic traits of the mutant differ from those of the wild type. Additionally, we observed the chloroplast structure of *spl43* using transmission electron microscopy to observe whether there were any changes, and we noted that its arrangement was sparser and exhibited more severe degradation compared with IR64 (Appendix A).

### 2.2. OsRPT5A Is a Constitutively Expressed Gene

To elucidate the spatiotemporal expression pattern of *OsRPT5A*, total RNA was extracted from various tissues at different developmental stages of IR64 and *spl43*, and its relative expression was quantified using quantitative reverse transcription-polymerase chain reaction (qRT-PCR). The results revealed that *OsRPT5A* is ubiquitously expressed across all examined tissues, including roots, stems, leaf sheaths, and panicles. For both the wild type and mutant, the lowest expression occurs in the root at 2 weeks, while the highest expression is in the leaves at 16 weeks (Figure 2A). Furthermore, a GUS reporter construct containing the *OsRPT5A* promoter was generated, and no staining was observed in the roots or stems. This means the GUS activity pattern corroborated the qRT-PCR findings, collectively indicating that *OsRPT5A* is constitutively expressed in rice (Figure 2B).

The correct production and localization of proteins are crucial for maintaining the functional operations of an organism. Therefore, studying the localization of proteins within the cell can provide preliminary insights into the function of the gene. To ascertain whether mutations in the OsRPT5A protein influence its subcellular localization, we expressed GFP-tagged OsRPT5A and its mutant proteins in rice protoplasts. The results revealed that both the OsRPT5A-GFP fusion protein and OsRPT5A^V318E^-GFP were localized in the cytoplasm, suggesting that the mutation in OsRPT5A does not alter its cellular localization (Figure 2C).

### 2.3. ROS Is Accumulated in spl43

*spl*-mutants are often associated with the hypersensitive response, a crucial mechanism for plant defense against pathogen infection, in which reactive oxygen species (ROS) play a significant role. Excessive ROS can lead to cell damage and death, resulting in disease lesions on the leaves. Additionally, although ROS are typically related to cell damage and defense responses, they may also be involved in cell signaling and the regulation of gene expression in certain contexts. Therefore, measuring ROS levels aids in comprehensively understanding the physiological and biochemical changes in the mutant. In this context, we evaluated physiological parameters in the *rpt5a* mutant and transgenic plants. These physiological parameters are among the measurable indicators of ROS homeostasis in plants. Our analysis demonstrated a significant elevation in H_2_O_2_ levels in *spl43* compared with the wild type, accompanied by a substantial reduction in catalase (CAT) activity, indicating a compromise in the ROS detoxification system. Additionally, the activities of superoxide dismutase (SOD) and peroxidase (POD) were markedly enhanced in *rpt5a*.

Conversely, the trends in soluble protein (SP), H_2_O_2_, and malondialdehyde (MDA) contents in the complemented plants closely resembled those in the IR64 wild type, signifying a partial restoration of ROS homeostasis in these plants (Figure 3A–F). The data from the knockout plants mirrored the mutant phenotype, confirming that the deletion of this gene triggers ROS accumulation (Figure 3G–L).

### 2.4. The Resistance of spl43 Is Enhanced

In the majority of rice spotted leaf mutants, alterations in resistance to pathogen infection are observed, accompanied by either upregulation or downregulation of defense response gene expression. Preliminary investigations have demonstrated that the *spl43* mutant exhibits enhanced resistance to *Xoo* race C2. To ascertain if this enhanced resistance is attributable to the *OsRPT5A* mutation, transgenic lines of *OsRPT5A* were inoculated with C2. The lesion lengths of the knockout lines were similar to those of the *spl43* mutant, suggesting that *OsRPT5A* plays a pivotal role in conferring resistance in rice (Figure 4A,B).

After inoculation, we examined the expression levels of some defense genes and found that they were upregulated in the knockout lines, further validating the inoculation results (Figure 4C).

## 3. Discussion

In our study, we successfully isolated the target gene responsible for the spotted leaf mutant phenotype through map-based cloning. In the F_2_ mapping population, there were 111 individuals with the mutant phenotype and 489 individuals with the wild-type phenotype. The observed segregation ratio does not conform to the expected 1:3 ratio (χ^2^ = 13.52, *p =* 3.84). This implies that the F_2_ population size used for mapping is not large enough or the mutation may link to unknown factors associated with abnormal segregation. Although the segregation of the positioning group does not comply with Mendel’s Law of Segregation, we validated its role in controlling this phenotype using transgenic lines. This indicates that the spotted leaf phenotype is controlled by a single recessive nuclear gene. The complemented lines effectively restored the wild-type phenotype, as the knockout lines exhibited spotted leaf phenotypes similar to the original mutant. This mutation induced significant differences in the major agronomic traits of the mutant compared with the wild type, highlighting its impact on plant development and yield. Notably, the chlorophyll content of the knockout lines remained similar to that of the mutant throughout the growth period. This suggests that the necrotic spots caused by the gene mutation adversely affected photosynthesis, thereby potentially impacting the plant’s energy production and overall health. Additionally, the knockout lines showed a reduction in 1000-grain weight and the seed setting rate, indicating a negative effect on reproductive success and yield and implying that the absence of the functional gene *RPT5A* led to abnormalities in plant agronomic traits.

The restoration of hydrogen peroxide levels to wild-type levels in the complemented lines suggests that the excessive accumulation of hydrogen peroxide in the mutant disrupts ROS scavenging system homeostasis and is closely related to the occurrence of spots. The high levels of hydrogen peroxide in the knockout lines enhance their resistance to C2, as confirmed by the high expression levels of defense genes 72 h after inoculation. Possible mechanisms include altering the strength of plant cell walls, activating other defense signaling pathways, or enhancing plant cell recognition of pathogens [9,10]. Abnormal RPT5A protein may interfere with normal plant growth and response pathways, leading to the formation of the spotted leaf phenotype.

*OsRPT5A* encodes a subunit of the 26S proteasome, playing a pivotal role in diverse plant species, particularly in leaf development, DNA damage repair, and responses to environmental stress. In rice, the *spl43* mutant demonstrates abnormal leaf development under zinc deficiency, highlighting the critical role of DNA damage mitigation in normal leaf development [11]. This suggests that RPT5A may underpin normal leaf morphology by engaging in DNA repair and ensuring genomic stability. In Arabidopsis, a mutation of *NAC103* alleviated DNA damage and cell death induced by excess boron in the root meristem of the *spl43* mutant, indicating that *RPT5A* might be instrumental in the plant’s response to environmental stress, such as excess boron, particularly in the DNA damage repair process [12]. Concurrently, both RPT2A and RPT5A are essential for zinc deficiency tolerance, suggesting that RPT5A is key in the plant’s adaptation to micronutrient deficiency, potentially by modulating protein degradation and signal transduction pathways [13]. In papaya, proteomic analysis suggests a link between RPT5A and typical sticky disease symptoms, implying that RPT5A may contribute to the plant’s disease response, possibly by regulating the degradation of immune-related proteins to participate in disease defense mechanisms [14]. Coupled with existing experimental evidence, we hypothesize that RPT5A may be regulated by upstream transcription factors, thereby playing a role in the regulation of ROS in plants, although the potential interacting transcription factors are still under investigation. Based on the STRING database (https://string-db.org/, accessed on 13 April 2023). we found that it interacts with protein kinases, indicating that RPT5A is not only tightly regulated upstream but also involved in the plant’s response to various stresses downstream through serine/threonine protein kinases, such as salt stress, drought stress, and low-temperature stress. The mutation site in the OsRPT5A protein is the 318th amino acid, where valine is mutated to glutamate, located in the SMART AAA domain. Given that valine is hydrophobic and glutamate is a negatively charged polar amino acid, many proteins with the AAA domain interact with other proteins through this domain. The mutation-induced change in the protein surface charge or hydrophobicity can impact protein–protein interactions, thereby affecting the assembly and stability of the protein complex and interfering with various cellular activities, including protein folding, membrane fusion, and DNA replication. In summary, the role of OsRPT5A in ROS regulation and plant resistance regulation remains an area for further investigation.

We confirmed the *OsRPT5A* gene’s role in controlling the spotted leaf phenotype through knockout and complementation experiments, demonstrating that this gene can be manipulated via genetic engineering to develop crop varieties with desired traits, such as improved disease resistance. Similar to many previously reported spotted leaf genes associated with protein ubiquitination, the OsRPT5A’s mutation—a component of the 26S proteasome complex—induces a spotted leaf phenotype in rice and enhances resistance to specific races of *Xoo*. However, whether it confers broad-spectrum resistance still requires confirmation. Further investigation into the precise role of this gene in disease defense could unveil novel mechanisms of resistance, offering new approaches to disease management. We observed that the mutation significantly impacts plant photosynthesis, growth, and yield, enhancing our understanding of how genes influence crucial agronomic traits through specific biological pathways. This insight is valuable for selecting or improving crop varieties with enhanced agronomic characteristics. As a newly identified spotted leaf mutant, the pathways and mechanisms underlying the development of the spotted leaves are not yet fully elucidated, but we hypothesize that critical involvement may occur upstream of the gene, regulated by a transcription factor.

## 4. Materials and Methods

### 4.1. Plant Materials and Growth Conditions

Ethyl methanesulfonate (EMS) was utilized to induce mutagenesis in the indica rice variety IR64, leading to the development of a comprehensive mutant library. Within this library, a specific leaf spot mutant, named as *spl43*, was isolated. Genetic analyses and gene mapping were performed on an F_2_ population derived from a cross between *spl43* and the japonica rice variety 80A90YR72. The cultivation of these experimental materials took place at the Fu Yang experimental base of the China National Rice Research Institute (CNRRI), under the auspices of the Chinese Academy of Agricultural Sciences. In May 2023, IR64, *spl43*, and transgenic lines were planted in experimental fields in Fuyang, Hangzhou. The planting spacing was 16.5 cm × 26.4 cm, with 8 rows and 10 plants per row. Harvesting was carried out at the end of September. Lighting, temperature, and humidity conditions were managed according to standard field practices.

### 4.2. The Construction of Vectors and the Acquisition of Transgenic Lines

For the complementation lines, the allelic gene *OsRPT5A* from the wild-type IR64, which includes a 3000 bp sequence upstream of the transcription start site, the complete 4319 bp genomic DNA, and a 2500 bp sequence downstream of the termination site, was cloned into the vector pCAMBIA1300, resulting in the construction of pCAMBIA1300-RPT5A. For the knockout lines, we employed CRISPR/Cas9 technology to excise specific genomic sequences, constructing the knockout vector prpt5a. Utilizing the GUS reporter vector pCambia1381Z, we cloned the 3000 bp promoter region of *OsRPT5A*, leading to the creation of the vector pCambia1381Z-RPT5A-GUS. For subcellular localization assays, the coding sequences of both the wild-type OsRPT5A and its variant, RPT5A^V318E^, were integrated into pYBA1132, yielding the vectors GFP-RPT5A and GFP-RPT5A^V318E^. All engineered constructs were transformed by BIORUN BIOSCIENCES CO., LTD, and the plant lines were cultivated at the Fu Yang experimental base of the China National Rice Research Institute, affiliated with the Chinese Academy of Agricultural Sciences.

### 4.3. Measurement of Photosynthetic Pigments

Photosynthetic pigments were extracted from the leaves of IR64, *spl43*, complementary lines, and knockout lines at the tillering stage. Leaves were cut into small segments, and 0.01 g of a sample was soaked in 1 mL of 95% ethanol in the dark for 48 h. Photosynthetic pigments were determined by measuring the absorbance at 470 nm (A_470_), 645nm (A_645_), and 663nm (A_663_) of the sample solution using a multifunctional microplate reader. The calculation formulas are as follows, where three replicates were performed for all measurements. The formulae for calculating chlorophyll and carotenoid (Car) content are as follows:Content of CHl a (mg/g) = 12.7 × A_663_ − 2.69 × A_645_; Content of CHl b (mg/g) = 22.9 × A_645_ − 4.68 × A_663_;
Content of Car (mg/g) = (1000 × A_470_ − 3.27 × Content of CHl a-104 × Content of CHl b)/229

The data were assessed by one-way ANOVA followed by Duncan’s multiple range test, with a significance threshold of *p* ≤ 0.05.

### 4.4. Agronomic Trait Evaluation

Three individual plants were randomly selected from each of the IR64, *spl43*, its complementation lines, and knockout lines to evaluate their agronomic traits at full maturity. These traits included panicle length, thousand-grain weight, and the seed setting rate. We used a ruler to measure the panicle length before harvest, and after harvesting, the seeds were dried, and the seed setting rate and thousand-grain weight were calculated. Three replicates were performed for all measurements. The data were assessed by one-way ANOVA followed by Duncan’s multiple range test, with a significance threshold of *p* ≤ 0.05.

### 4.5. Measurement of Physiological Indexes

According to the protocols provided by the Nanjing JianCheng Bioengineering Institute’s assay kits(Nanjing Jiancheng Bioengineering Institute, Nanjing, China), leaves of IR64, *spl43*, its complementation lines, and knockout lines at the tillering stage were homogenized to prepare a uniform slurry. The kits used for the measurement are as follows: Total protein quantitative assay kit (Nanjing Jiancheng Bioengineering Institute, Nanjing, China, A045-2-2), Malondialdehyde (MDA) assay kit (TBA method) (Nanjing Jiancheng Bioengineering Institute, Nanjing, China, A003-1-2), Hydrogen Peroxide assay kit (A064-1-1), Catalase (CAT) assay kit (Visible light) (Nanjing Jiancheng Bioengineering Institute, Nanjing, China, A007-1-1), Peroxidase assay kit (Nanjing Jiancheng Bioengineering Institute, Nanjing, China, A084-3-1), and Total Superoxide Dismutase (T-sod) assay kit (Hydroxylamine method) (Nanjing Jiancheng Bioengineering Institute, Nanjing, China, A001-1-2). All information about the kits can be found on their official website (http://www.njjcbio.com/, accessed on 15 August 2023). The concentrations of stress biomarkers including soluble proteins (SPs), malondialdehyde (MDA), and hydrogen peroxide (H_2_O_2_), as well as the enzymatic activities of superoxide dismutase (SOD), peroxidase (POD), and catalase (CAT), were quantitatively assessed. Three replicates were performed for all measurements. The data were assessed by one-way ANOVA followed by Duncan’s multiple range test, with a significance threshold of *p* ≤ 0.05.

### 4.6. Subcellular Localization

To isolate protoplasts, start by selecting 3-week-old etiolated rice leaves, which should be thoroughly washed with distilled water and then dried. Next, cut the leaves into strips measuring 0.5–1 mm and immerse them in an enzyme solution composed of Cellulase R10, Macerozyme R10, 0.6M mannitol, 10 mM MES buffer (pH 5.7), 10 mM calcium chloride, and 0.1% BSA, ensuring complete submersion. Apply vacuum treatment for 15–30 min to remove air bubbles and enhance penetration of the enzyme solution. Incubate the leaves in the dark at 25–28 °C with gentle shaking for 3–6 h. After incubation, filter the enzymatically digested leaf mixture through double-layer gauze and collect the filtrate. Centrifuge this filtrate at room temperature (100× *g* for 5 min) to pellet the protoplasts, carefully discarding the supernatant and resuspending the pellet in a washing buffer. Repeat the centrifugation and washing process 2–3 times to thoroughly remove residual enzymes. For the final step, after the last centrifugation, discard the supernatant and resuspend the protoplasts in MMG buffer (0.4 m mannitol, 15 mM MgCl_2_, 4 mM MES, pH 5.7). Under sterile conditions, transfer 100 µL of the protoplast suspension into a 1.5 mL centrifuge tube. Introduce 10–20 µg of plasmid DNA into the suspension and mix gently. Add 110 µL of PEG solution, previously diluted to 20% with MMG buffer, and mix gently. Allow the mixture to incubate at room temperature for 5 min before adding 440 µL of transfection buffer, mixing gently again to further dilute the PEG. After an additional 10 min of incubation at room temperature, add 1 mL of MMG buffer and mix gently. Centrifuge at low speed (100× *g* for 2 min), discard the supernatant, and resuspend the protoplast pellet in fresh MMG buffer. Cover the container with a lid, wrap it in aluminum foil, and incubate in the dark for 16–24 h at approximately 25 °C. Finally, use a confocal microscope to observe the fluorescence signal in the protoplasts.

### 4.7. Gene Mapping

DNA was extracted utilizing the Cetyltrimethylammonium bromide (CTAB) method [15]. Rice genomic sequences were retrieved from the GRAMENE database (http://ensembl.gramene.org/, accessed on 15 July 2021) and the Chinese Rice Genome Database (http://rice.genomics.org.cn/, accessed on 15 July 2021). Primers were specifically designed to amplify genomic regions based on the comparative genomic analysis between the japonica variety “Nipponbare” and the indica variety “9311”. Primer designs and sequence analyses were facilitated by utilizing resources from the GRAMENE and MSU Rice Genome Annotation Project databases (http://rice.uga.edu/, accessed on 22 July 2021). PCR amplifications were performed according to the protocols specified by Vazyme (P222-03).

### 4.8. qRT-PCR Analysis

RNA was extracted using the Invitrogen TRIzol reagent (ThermoFisher, Waltham, MA, USA) according to the manufacturer’s instructions. To assess the impact of inoculation with *Xanthomonas oryzae* pv. *oryzae* on the expression levels of defense genes, RNA was isolated from the leaves of the IR64, *spl43*, CR-1, and CR-2 lines at 72 h post-inoculation. The qRT-PCR analyses were conducted using the SYBR Green qPCR Master Mix from Vazyme (Q511-02). The rice ubiquitin gene (*LOC_Os03g13170*) served as an internal control. Data analysis was performed using the 2^−ΔΔCt^ method, with each experimental condition replicated three times. The primers utilized for qRT-PCR are detailed in Appendix A.

### 4.9. Inoculation with the Bacterial Blight Pathogen

The method detailed in [16] was conducted on the tillering-stage leaves of the IR64, *spl43*, CR-1, and CR-2 lines using the race C2 of *Xanthomonas oryzae* pv. *oryzae*. Lesion lengths were measured 14 days after inoculation. For each group, six fully expanded leaves from different plants were inoculated. Six replicates were measured to analyze the data. The data were assessed by one-way ANOVA followed by Duncan’s multiple range test, with a significance threshold of *p* ≤ 0.05.

### 4.10. Chloroplast Structure Observation

For the TEM observation of chloroplast structure, the leaves of IR64 and *spl43* at the tillering stage were first soaked in 2.5% glutaraldehyde for 24 h. The samples were processed at the Instrument and Equipment Sharing Service Technology Platform of CNRRI. Initially, each sample underwent three successive rinses with 0.1M phosphate buffer, each lasting 15 min, to ensure the removal of extraneous materials. Following this, the samples were fixed in a 1% osmium tetroxide solution for 2–3 h to stabilize cellular structures and then rinsed again three times in 0.1M phosphate buffer for 15 min each. The dehydration sequence involved a series of ethanol treatments under refrigerated conditions at 4 °C as follows: starting with 50% ethanol, increasing to 70%, and finally, 90%, with each step lasting 15–20 min. This was followed by a mixture of 90% ethanol and 90% acetone (1:1 ratio) for 15–20 min and then 90% acetone alone for an additional 15–20 min. Complete dehydration was achieved by treating the samples with 100% acetone at room temperature, repeated three times for 15–20 min each. For embedding, the samples were first immersed in a solution of pure acetone and embedding resin at a 2:1 ratio for 3–4 h at room temperature. They were then transferred to a 1:2 ratio of pure acetone and embedding resin, where they remained overnight at room temperature. The samples were subsequently placed in pure embedding resin at 37 °C for 2–3 h. Polymerization was conducted in phases as follows: initially at 37 °C overnight in an oven, followed by 12 h at 45 °C, and finalized at 60 °C for 24 h to ensure complete resin hardening. The embedded samples were then sectioned into 50–60 nm ultrathin slices using an ultramicrotome. These sections were double-stained with 3% uranyl acetate and lead citrate to enhance contrast and then examined and imaged using TEM.

## 5. Conclusions

The spl-mutant *spl43* is characterized by stable inherited reddish-brown spotted leaves. We used map-based cloning to identify the genetic basis of this phenotype and named the target gene *LOC_Os02g56000* as *OsRPT5A*. It shows a substitution of valine (Val) for glutamic acid (Glu) in the protein. The complementation lines successfully restored the phenotypes, and the knockout lines exhibited the spotted leaf phenotype, indicating that the defect of *OsRPT5A* can lead to the spotted leaf phenotype with enhanced resistance to *Xoo*. *OsRPT5A* is a constitutively expressed gene, its mutation results in a decrease in photosynthetic pigment content, the accumulation of reactive oxygen species, and reduced agronomic traits.

## Figures and Tables

**Figure 1 ijms-25-06637-f001:**
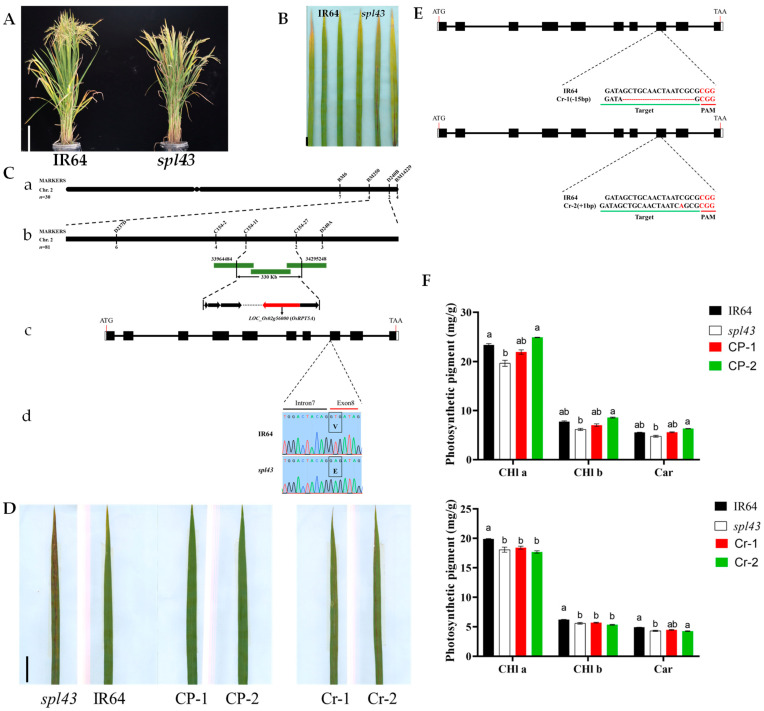
*spl43* and transgenic lines. (**A**) Phenotypes of IR64 and *spl43* at the maturity stage. Bar = 20 cm. (**B**) Leaf phenotypes of IR64 and *spl43* at the maturity stage. Bar = 2 cm. (**C**) Map-based cloning of *spl43*. (**C**-**a**) The locus to an interval was confined between markers RM6 and RM14229 on chromosome 2. (**C**-**b**) The locus to an interval was finally confined between markers C154-11 and C154-27. (**C**-**c**) The target gene is *LOC_Os02g56000*. (**C**-**d**) A point mutation from thymine (T) to adenine (A) occurred in the eighth exon of OsRPT5A, leading to the substitution of valine (V) for glutamic acid (E). (**D**) Leaf phenotypes of IR64, *spl43*, and complementary lines (CP-1, CP-2) and knockout lines (Cr-1, Cr-2). Bar = 3.8 cm. (**E**) CRISPR/Cas9-mediated mutations at the target sites in knockout lines (Cr-1, Cr-2); small red letters indicate the corresponding base insertions and dot lines indicate deletions. Bar = 3.75 cm. (**F**) Photosynthetic pigment. Different letters signify statistically significant differences assessed by one-way ANOVA followed by Duncan’s multiple range test, with a significance threshold of *p* ≤ 0.05. Error bars represent SD (*n* = 3).

**Figure 2 ijms-25-06637-f002:**
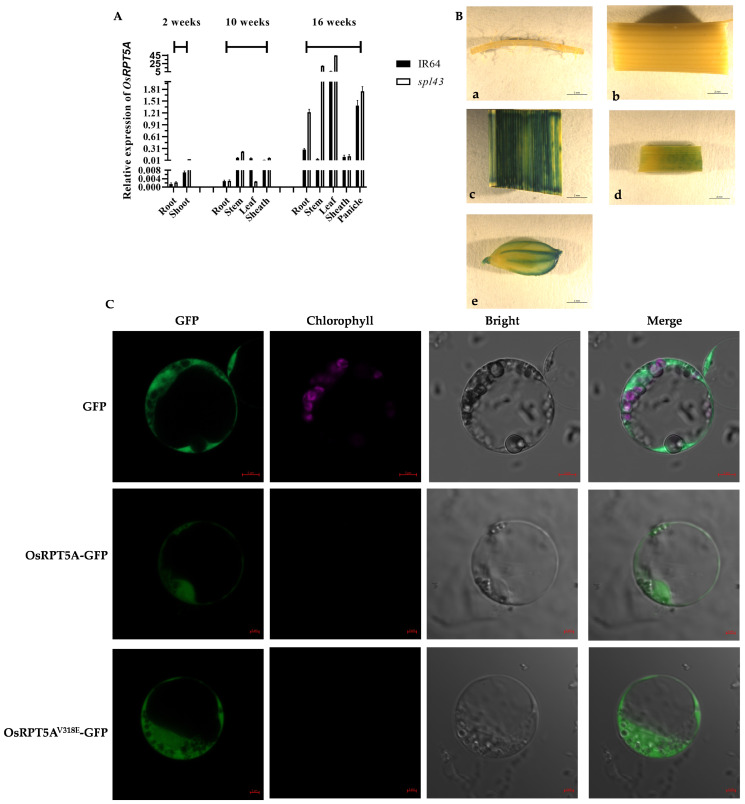
The spatiotemporal expression pattern analysis and subcellular localization of *spl43*. (**A**) At 2 weeks, 10 weeks, and 16 weeks, the relative expression levels of *OsRPT5A* in the roots, stems, leaves, leaf sheaths, and panicles of IR64 and *spl43*. Error bars represent SD (*n* = 3). (**B**) GUS staining analysis. (**B**-**a**) Root; (**B**-**b**) stem; (**B**-**c**) leaf; (**B**-**d**) leaf sheath; and (**B**-**e**) seed. Bar = 2 mm; (**C**) The subcellular location of *OsRPT5A*. Bar = 5 mm. Error bars represent SD (*n* = 3).

**Figure 3 ijms-25-06637-f003:**
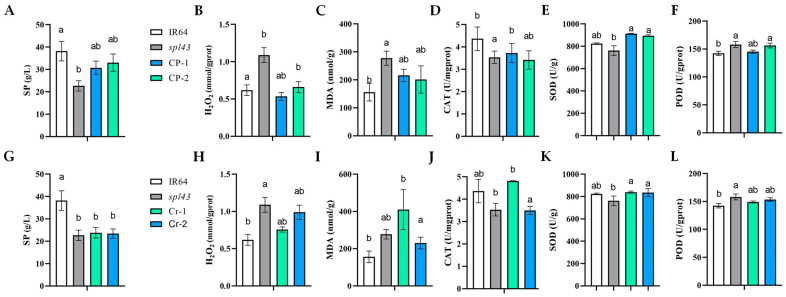
The physiological parameters of *spl43* and transgenic lines. (**A**,**G**) SP content; (**B**,**H**) H_2_O_2_ content; (**C**,**I**) MDA content; (**D**,**J**) CAT activity; (**E**,**K**) SOD activity; and (**F**,**L**) POD activity. Different letters signify statistically significant differences, as assessed by one-way ANOVA followed by Duncan’s multiple range test, with a significance threshold of *p* ≤ 0.05. Error bars represent SD (*n* = 3).

**Figure 4 ijms-25-06637-f004:**
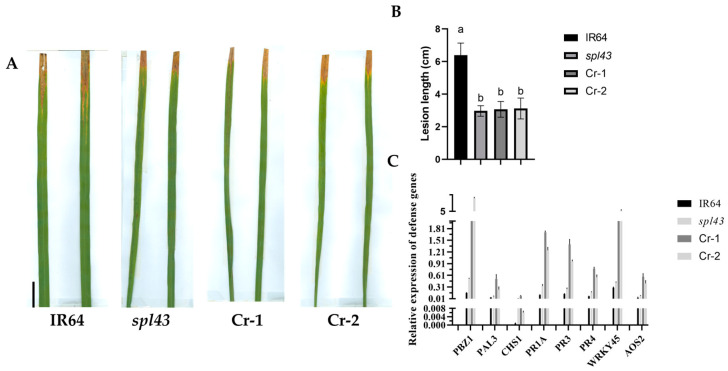
Evaluation of disease resistance and relative expression of defense genes. (**A**) Results of inoculation of IR64, *spl43*, and the knockout lines with *Xoo* race C2 at the tillering stage. Bar = 3 cm. (**B**) Lesion length. (**C**) Expression levels of genes after inoculation. Different letters signify statistically significant differences, as assessed by one-way ANOVA followed by Duncan’s multiple range test, with a significance threshold of *p* ≤ 0.05. Error bars represent SD (*n* = 3).

## Data Availability

The data are contained in this article or the Appendix A.

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
