# Peer review of "The Identification and Gene Mapping of Spotted Leaf Mutant *spl43* in Rice"

_ijms, 2024, doi:10.3390/ijms25126637_

Round 1
Reviewer 1 Report
Comments and Suggestions for Authors
The manuscript “OsPRT5A: A Key Regulator of Enhanced Immune Responses in Rice Through Loss-of-Function Mutation” identified OsRPT5A as the causal gene of lesion mimic mutant of rice. The non-synonymous mutation in the mutant does not alter the subcellular localization of the protein, thus it was suggested that the mutation altered the protein-protein interaction. The mutation mainly affects the ROS balance, enhance defense marker genes expression, limited the leaf lesion length upon infection. The major drawback of this manuscript is not well written that a lot of information is missing. Thus, it is not suggested to accept the manuscript in the current form.
The basic information for the mapping is basically missing in this manuscript. How many F2 individuals were used? How phenotyping of these F2 individuals was done? Was the phenotype quantitatively measured or qualitatively measured? How was the phenotype distributed. What are the markers used? How many of them? What is the distribution of the markers? How the fine mapping was done? Any other information needed for the understanding of how the mapping was done should be included.
Although the title claimed that OsPRT5A is a key regulator for immune response, there is only one short section in this manuscript investigating the immune response of the OsPRT5A mutant. The prt5a was also isolated as a lesion mimic mutant that its immune related function was a coincidence, although it is known that lesion mimic mutants usually have high resistance to pathogen infection. Therefore, the manuscript should be revised substantially to fit the title, or the tile should be revised to fit the content of the manuscript.
The rationale for doing each experiment was not clearly stated. For example, it is largely unclear that why chlorophyll content was measured. It is also largely unknown that why subcellular localization study was done as there is no indication that the non-synonymous mutation will affect the subcellular localization of the protein. This also applies to other experiments in the manuscript.
Please explain why fresh weight was used as the denominator for chlorophyll measurement while protein content was used as the denominator for ROS and ROS scavenging enzyme measurements. If fresh weight was used as the denominator for ROS measurements, the results would be completely different.
The authors mentioned that “the segregation of the positioning group does not comply with Mendel’s Law of Segregation”. Data should be shown, and explanation supported by experiment should be given.
In figure 1D, the lesion of the two Cr lines was not as obvious as the rpt5a mutant. Can the authors explain about it?
For the original mutant, before the mapping and all the genetic studies, the causal gene should be unknown. In such case, it should not be named rpt5a (line 72) before the gene was identified.
Figure 1F. The error bars were so tight that they do not look like the standard deviation of biological replicates. Please confirm the replicates in all the experiments are real biological replicates as indicated in the materials and methods.
The relationship between NAC103 and RPT5a was not clearly stated.
Line 219. Do not mention about unpublished data or otherwise please show the real data as the data is not peer-reviewed which can cause confusion in the future.
Materials and methods for subcellular localization is missing.
Materials and methods for chlorophyll measurement is missing.
More information for “4.4 Measurement of physiological indexes” should be added. What are the kits used? Is there any reference for the measurement methods?
The full name of CTAB should be Cetyltrimethylammonium bromide.
How samples were examined in the “Instrument and Equipment Sharing Service Technology Platform of CNRRI” should be detailed.
Comments on the Quality of English Language
NA
Reviewer 2 Report
Comments and Suggestions for Authors
Dear Authors,
Correction suggestions are described in the text.
Best regards,

Dear Authors,
Correction suggestions are described in the text.
Best regards,
Round 2
Reviewer 1 Report
Comments and Suggestions for Authors
The manuscript has been substantially revised. However, some clarifications are still needed.
Please clarify the number of F2 individuals used. In cover letter, the authors said 500 individuals were used. Yet 489 wild type and 111 mutant like individuals (489+111=600) were found. Which one is correct?
On the other hand, 30 individuals with MUTANT phenotype were used for initial mapping. 80 individuals with MUTANT phenotype were used for fine mapping. What about the wildtype like individuals?
The primers, sequence, and physical locations for the 200 SSR markers and additional markers, especially those designed by the authors of this work, should be included in the supplementary of the manuscript.
Please carry out Chi Square test to confirm the 3:1 ratio and revise the conclusion if necessary.
The additional information was not coherent with the old content. For example, Line135. The newly added content is explaining the inconsistency of the phenotype between mutant and Cr lines. However, the inconsistency has not yet been mentioned anywhere in the manuscript. Some of the newly added information has the same issue.
The added rationale does not explain well why the experiment should be done. It seems that the authors do not understand what they are doing. FOR EXAMPLE, chlorophyll content is not a good parameter for leaf development. Measurement of ROS may not be related to cell signaling, but normally related to hypersensitive response as the cause of leaf lesion.
The manuscript required substantial type editing especially the newly added content.
Comments on the Quality of English LanguageType editing of the English is required especially the M&M section.
Round 3
Reviewer 1 Report
Comments and Suggestions for Authors
The manuscript has been substantially revised. I really appreciate the authors' working in making this manuscript. However, there are two minor issues that I would like the authors to address.
I am not working in rice research, and I won't be able to re-run all the mapping results as part of the reviewing process. However, the SSR information (Primer sequences and physical location) is the foundation of this research. This will also be essential for future validation of the results by other researchers in the rice research field. The information should be ready, and it won't be hard to prepare. I see not point of not including them as supplementary information in this manuscript. The information MUST be included.
The statement "This could be due to mutations at different sites having varying impacts on protein function, leading to different levels of disruption in the plant's reactive oxygen species (ROS) homeostasis, and thus resulting in phenotypes that are less pronounced than those of the mutant." is explaining the discrepancy between the phenotype of the mutant line and the Cr lines. The authors have not really mentioned about this issue in the manuscript.
